# Ion Exchange of One-Pot Synthesized Cu-SAPO-44 with $NH_4NO_3$ to Promote Cu Dispersion and Activity for Selective Catalytic Reduction of $NO_x$ with $NH_3$

**Nana Zhang [1], Ying Xin [1,\*] , Qian Li [1], Xicheng Ma [2], Yongxin Qi [3], Lirong Zheng [4] and Zhaoliang Zhang [1,\*]**

[1] School of Chemistry and Chemical Engineering, Shandong Provincial Key Laboratory of Fluorine Chemistry and Chemical Materials, University of Jinan, Jinan 250022, China; chm_zhangnn@mail.ujn.edu.cn (N.Z.); chm_liqian@ujn.edu.cn (Q.L.)

[2] School of Chemistry and Chemical Engineering, Shandong University, Jinan 250100, China; maxch@sdu.edu.cn

[3] School of Material Science and Technology, Shandong University, Jinan 250100, China; qyx66@sdu.edu.cn

[4] Institute of High Energy Physics, Chinese Academy of Sciences, Beijing 100049, China; zhenglr@ihep.ac.cn

\* Correspondence: chm_xiny@ujn.edu.cn (Y.X.); chm_zhangzl@ujn.edu.cn (Z.Z.); Tel.: +86-0531-8973-6032 (Y.X. & Z.Z.)

**Abstract:** Cu-containing CHA type (Cu-CHA) zeolites have been widely investigated owing to their excellent low-temperature activity and high hydrothermal stability in selective catalytic reduction of $NO_x$ with $NH_3$ ($NH_3$-SCR). Herein, a series of Cu-SAPO-44 zeolites were prepared by one-pot method with dual-amine templates and the subsequent ion exchange (IE) with $NH_4NO_3$. The effect of $NH_4NO_3$ treatment on Cu species was investigated by X-ray powder diffraction (XRD), $N_2$ adsorption-desorption isotherm, inductively coupled plasma (ICP); field-emission scanning electron microscope (FE-SEM), high-resolution transmission electron microscope (HRTEM), X-ray absorption fine structure (XAFS), and $H_2$-temperature-programmed reduction ($H_2$-TPR). The results indicated that—besides the main SAPO-44 structure—the CuO phase was detected by XRD in original samples. After IE with $NH_4NO_3$, the Cu contents decreased greatly from ICP analysis. The removal of CuO agglomerations and the presence of highly dispersed CuO nanoparticles (~2.36 nm) were confirmed by SEM, TEM and $H_2$-TPR. Furthermore, a significant increase in the proportion of isolated $Cu^{2+}$ was derived from XAFS. As a result, the activity at higher temperature (≥350 °C) was improved a lot.

**Keywords:** selective catalytic reduction; nitrogen oxide; one-pot synthesis; Cu-SAPO-44; ion exchange; $NH_4NO_3$

## 1. Introduction

Aqueous ion exchange (IE) is the most widely used method in preparation of small pore Cu-chabazite (Cu-CHA) zeolites [1–3]. Multiple IE procedures are time consuming; therefore, one-pot synthesis methods were hence developed [4–9]. For instance, Ren et al. pioneered the synthesis of Cu-SSZ-13 using $Cu^{2+}$-tetraethylenepentamine (Cu-TEPA) complex as structure-directing agent (SDA), with Cu loading in the range 0–10 wt.% [10]. Soon after, Cu-SAPO-34 zeolites with controllable Cu-loadings were synthesized by one-pot method using Cu-TEPA and low-cost SDAs [6,11–13]. More recently, another Cu-CHA zeolite, Cu-SAPO-44, was also reported [14]. In contrast to IE method, one-pot method could directly introduce $Cu^{2+}$ ions into zeolites frameworks, and thus achieve high Cu loading and high dispersion of Cu species, leading to excellent $NH_3$-SCR activity [15].

The downside, however, is that one-pot synthesis method tend to introduce excessive Cu species into CHA zeolites, which is detrimental to the hydrothermal stability of Cu-CHA catalysts [5]. IE with $NH_4NO_3$ was discovered as an efficient approach to achieve moderate Cu-loadings in one-pot synthesis of Cu-CHA zeolites by removing the excessive Cu species [10,16,17]. Xie et al. reported that one-time IE with $NH_4NO_3$ was completely sufficient to eliminate excess Cu species from the SSZ-13 structure and re-disperse the remaining $Cu^{2+}$ ions [15]. Guo et al. demonstrated that $Cu^{2+}$ ions migrated from large cages to more stable locations in six-membered rings of CHA structure after IE with $NH_4NO_3$ [17]. Nevertheless, there still lacks direct observation on the structural change of Cu species from one-pot synthesized Cu-CHA zeolites.

Herein, a series of Cu-SAPO-44 zeolites were fabricated through one-pot method with the dual-amine templates (denoted as $Cu_x$-SAPO-44) followed with ion exchange with $NH_4NO_3$ solution (denoted as $Cu_x$-SAPO-44-IE). The effects of IE with $NH_4NO_3$ on the textural and physical-chemical properties of Cu-SAPO-44, as well as the nature and location of active Cu species, were investigated. Interestingly, IE with $NH_4NO_3$ removed excessive CuO agglomerations, leading to a high proportion of isolated $Cu^{2+}$ ions responsible for the high SCR activity.

## 2. Results and Discussion

### 2.1. XRD

Figure 1 displays XRD patterns of Cu-SAPO-44 zeolites before and after IE with $NH_4NO_3$. The diffraction peaks for all samples are in accordance with those of pure SAPO-44 reported in our previous work [18], confirming the CHA structure [14,19]. However, the CuO phase was present in $Cu_x$-SAPO-44. Interestingly, the $Cu_x$-SAPO-44-IE samples show only diffraction patterns of SAPO-44, suggesting that IE with $NH_4NO_3$ can remove the excess CuO species and simultaneously retain the SAPO-44 structure.

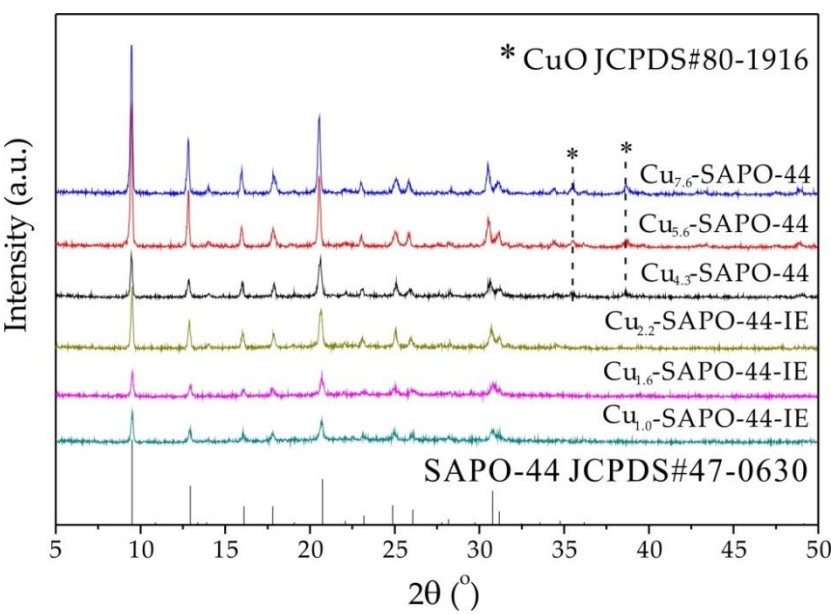

**Figure 1.** XRD patterns of $Cu_x$-SAPO-44 and $Cu_x$-SAPO-44-IE.

### 2.2. ICP

The compositions of all samples were analyzed by ICP and listed in Table 1. Although the Si/Al ratios do not change much after $NH_4NO_3$ treatment, the Cu contents decrease evidently after IE with $NH_4NO_3$, in accordance with above XRD results.

**Table 1.** ICP results for Cu$_x$-SAPO-44 and Cu$_x$-SAPO-44-IE.

| Sample | Element Content from ICP (wt.%) | | | | |
|---|---|---|---|---|---|
| | **Cu** | **Si** | **Al** | **Si/Al** | **P** |
| Cu$_{4.3}$-SAPO-44 | 4.3 | 5.7 | 17.2 | 0.32 | 16.5 |
| Cu$_{5.6}$-SAPO-44 | 5.6 | 6.4 | 16.7 | 0.37 | 13.6 |
| Cu$_{7.6}$-SAPO-44 | 7.6 | 5.1 | 17.6 | 0.28 | 15.1 |
| Cu$_{1.0}$-SAPO-44-IE | 1.0 | 6.1 | 18.8 | 0.31 | 18.2 |
| Cu$_{1.6}$-SAPO-44-IE | 1.6 | 7.1 | 18.7 | 0.37 | 14.0 |
| Cu$_{2.2}$-SAPO-44-IE | 2.2 | 4.8 | 20.2 | 0.23 | 18.4 |

*2.3. Catalytic Activity*

SCR performances of Cu$_x$-SAPO-44 and Cu$_x$-SAPO-44-IE are shown in Figure 2. The Cu$_x$-SAPO-44 samples show 90% NO$_x$ conversion between 200 and 350 °C as well as high N$_2$ selectivity. However, NO$_x$ conversions decrease rapidly when the reaction temperatures are above 300 °C, due to the over-oxidation of NH$_3$ in the presence of CuO [15,20,21]. Notably, Cu$_x$-SAPO-44-IE samples exhibit improved NO$_x$ conversion at high temperatures, reaching above 90% at 200–450 °C with nearly 100% N$_2$ selectivity. This is consistent with the fact that the excess CuO species promote the oxidation of NH$_3$ [10,15,17]. To gain more information about the beneficial role of IE, further characterization was performed taking the control sample (Cu$_{5.6}$-SAPO-44) and the best sample (Cu$_{1.6}$-SAPO-44-IE) as examples.

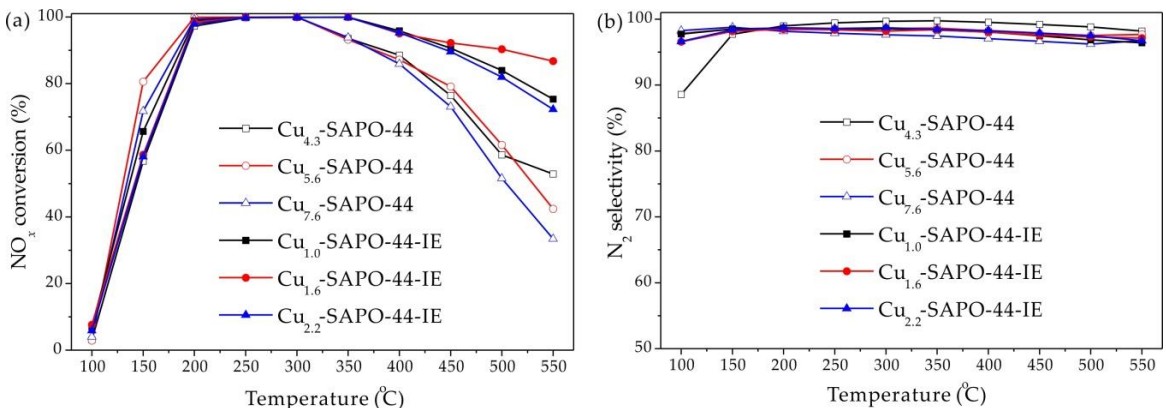

**Figure 2.** NO conversion (**a**) and N$_2$ selectivity (**b**) for Cu$_x$-SAPO-44 and Cu$_x$-SAPO-44-IE as a function of temperature. Reaction conditions: [O$_2$] = 5.3 vol.%, [NO] = [NH$_3$] = 500 ppm, balance He, total flow rate = 300 mL/min, GHSV = 100,000 h$^{-1}$.

Based on the practical consideration, NH$_3$-SCR performance of the Cu$_{1.6}$-SAPO-44-IE catalyst was also evaluated as 5 vol.% H$_2$O contained in feeding gas under the GHSV of 100,000 h$^{-1}$, and the result was shown in Figure 3. Unsurprisingly, a decrease in NO$_x$ conversion was observed at low temperatures (<250 °C), which could be ascribed to the competitive adsorption of H$_2$O with NH$_3$. In contrast, NO$_x$ conversion at high temperatures (>400 °C) was promoted, most likely due to the inhibition effect of H$_2$O on the unselective catalytic oxidation of NH$_3$ [22].

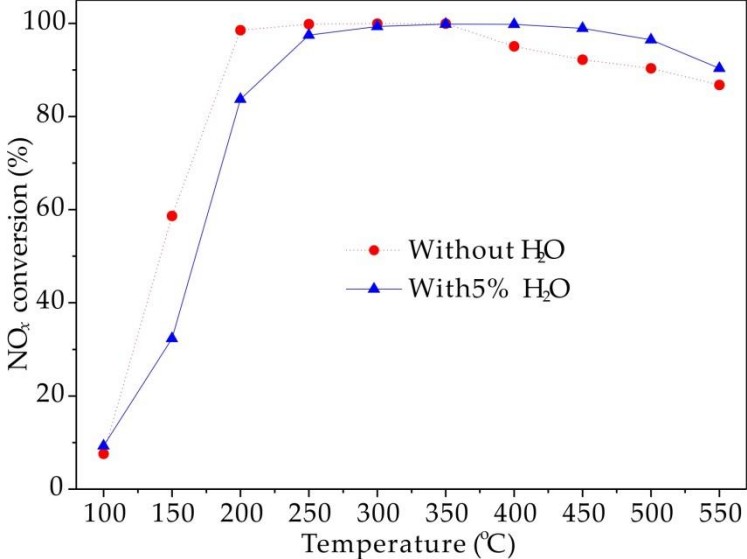

**Figure 3.** Effect of $H_2O$ on $NH_3$-SCR performance of $Cu_{1.6}$-SAPO-44-IE.

### 2.4. $N_2$ Adsorption/Desorption

The $N_2$ adsorption/desorption isotherms of $Cu_{5.6}$-SAPO-44 and $Cu_{1.6}$-SAPO-44-IE exhibit the typical microporosity (Figure 4), similar to the pure SAPO-44 [18]. As a result, their BET surface areas are comparable. Furthermore, the pore volume of $Cu_{1.6}$-SAPO-44-IE slightly increased in comparison with $Cu_{5.6}$-SAPO-44 (Table 2), suggesting the CuO species that block the pores in CHA zeolite have been partially removed, as detected by ICP (Table 1).

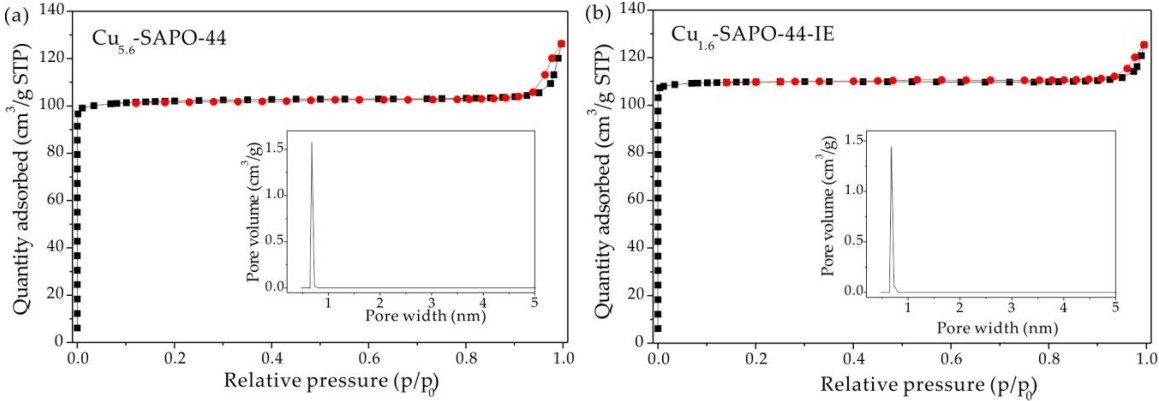

**Figure 4.** $N_2$ adsorption/desorption isotherms and pore diameter distribution for (**a**) $Cu_{5.6}$-SAPO-44 and (**b**) $Cu_{1.6}$-SAPO-44-IE.

**Table 2.** Textural properties and XAFS fitting results for $Cu_{5.6}$-SAPO-44 and $Cu_{1.6}$-SAPO-44-IE.

| Sample | BET Surface Areas (m²/g) | Pore Volume (cm³/g) | $Cu^{2+}$/CuO Data from LCF of XANES Spectra (wt.%/wt.%) |
|---|---|---|---|
| $Cu_{5.6}$-SAPO-44 | 338.8 | 0.15 | 22.4/77.6 |
| $Cu_{1.6}$-SAPO-44-IE | 363.7 | 0.17 | 69.7/30.3 |

### 2.5. Electron Micrograph

SEM and TEM were conducted for $Cu_{5.6}$-SAPO-44 and $Cu_{1.6}$-SAPO-44-IE (Figure 5). It is notable that both $Cu_{5.6}$-SAPO-44 (Figure 5a,b) and $Cu_{1.6}$-SAPO-44-IE (Figure 5d,e) show the cubic-like rhombohedral morphology, similar to those reported in literature [19]. According to EDS elemental

mappings, some large Cu-containing agglomerations are observed in $Cu_{5.6}$-SAPO-44 that can be ascribed to the aggregated CuO nanoparticles (Figure 5c), owing to its high Cu content. As reported by He et al., the excess Cu species in Cu-SSZ-13 zeolites could be effectively removed by ion exchange using $NH_4NO_3$ solution [15]. Accordingly, large CuO particles can be scoured into CuO nanocrystals and re-dispersed as $Cu^{2+}$ ions during this ion exchange process. In this work, after IE with $NH_4NO_3$, highly dispersed Cu-containing species were observed over the entire zeolite as for $Cu_{1.6}$-SAPO-44-IE (Figure 5f), which can be attributed to the remaining CuO nanoparticles and $Cu^{2+}$ ions at the framework ion-exchange sites. In order to clarify the highly dispersed Cu-containing species after IE with $NH_4NO_3$, $Cu_{1.6}$-SAPO-44-IE was further characterized by TEM (Figure 5g–i). The well-dispersed nanocrystals were observed in $Cu_{1.6}$-SAPO-44-IE (Figure 5g), which were proved to be CuO nanoparticles by identifying the characteristic spacings of 2.27 and 1.87 Å for the (2 0 0) and the (-2 0 2) lattice planes of monoclinic CuO (Figure 5h). A particle count taken from many TEM images, obtained from different regions of the sample, confirmed the presence of monodispersed CuO nanoparticles with a mean diameter of 2.36 nm anchored on the surface of $Cu_{1.6}$-SAPO-44-IE (Figure 5i). Combined with ICP data, the results confirm that IE with $NH_4NO_3$ can not only remove the excessive Cu species (in the form of CuO agglomerations) but also improve the dispersion of the remaining CuO nanoparticles. This result is consistent with the observation from other researchers that CuO aggregates decrease the SCR performance at high temperature range as a result of parasitic $NH_3$ oxidation [15,23].

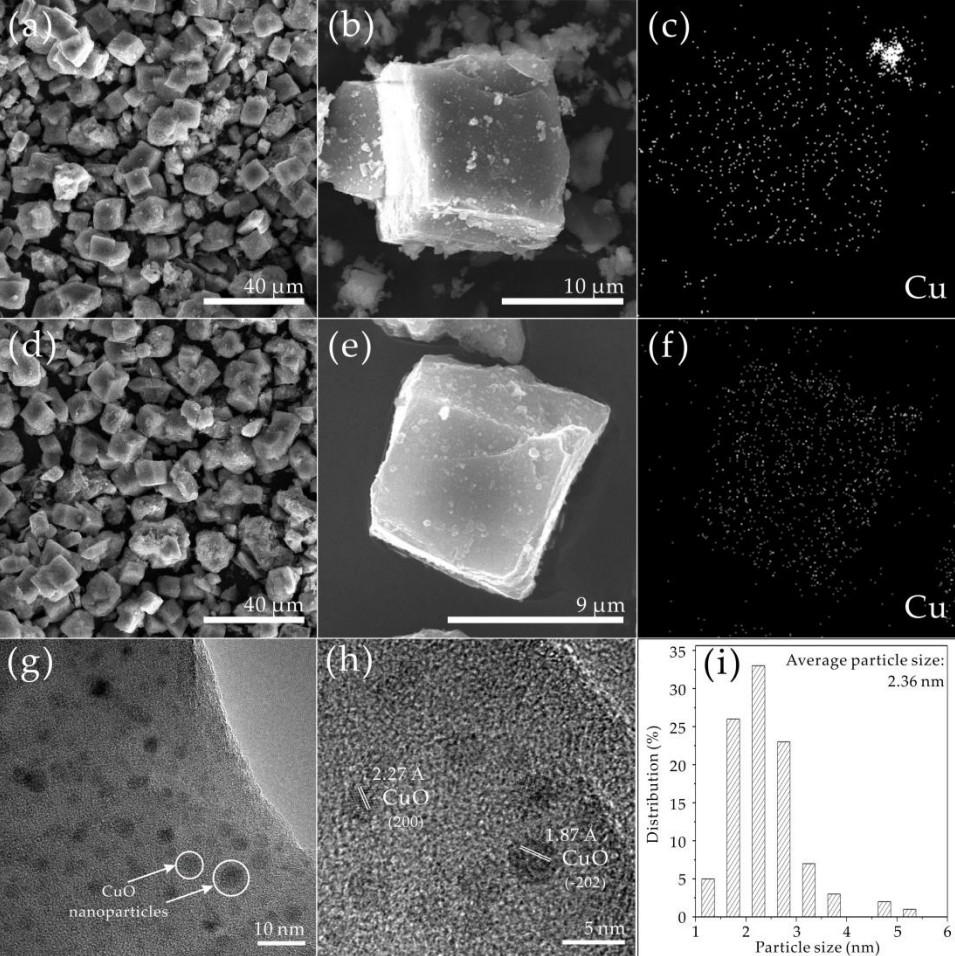

**Figure 5.** Electron micrograph of $Cu_{5.6}$-SAPO-44 and $Cu_{1.6}$-SAPO-44-IE. FESEM images for $Cu_{5.6}$-SAPO-44 (**a**,**b**) and $Cu_{1.6}$-SAPO-44-IE (**d**,**e**), Cu mapping for $Cu_{5.6}$-SAPO-44 (**c**) and $Cu_{1.6}$-SAPO-44-IE (**f**); TEM image (**g**), HRTEM image (**h**) and particle size distribution of CuO nanoparticles (**i**) of $Cu_{1.6}$-SAPO-44-IE.

## 2.6. H$_2$-TPR

H$_2$-TPR was also performed on Cu$_{5.6}$-SAPO-44 and Cu$_{1.6}$-SAPO-44-IE, and the results were shown in Figure 6. For Cu$_{5.6}$-SAPO-44 (a), the obvious reduction peak centered at ~300 °C could be ascribed to the reduction of CuO (>3 nm) to Cu$^0$ [24]. However, this peak disappeared in Cu$_{1.6}$-SAPO-44-IE (b), and a broad reduction peak ranging from 160 to 750 °C emerged, suggesting the removal of large agglomerated CuO particles and the existence of highly dispersed Cu$^{2+}$ and CuO microcrystals after NH$_4$NO$_3$ treatment, in accordance with the results of XRD and SEM.

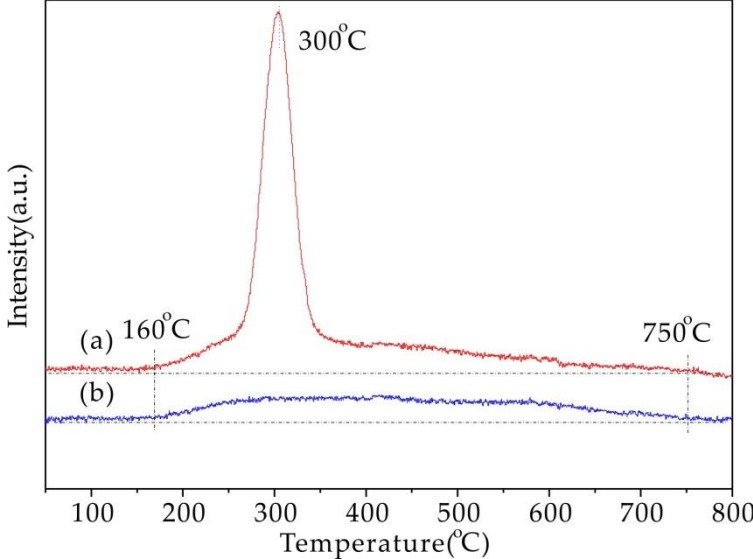

**Figure 6.** H$_2$-TPR profiles of Cu$_{5.6}$-SAPO-44 (**a**) and Cu$_{1.6}$-SAPO-44-IE (**b**).

## 2.7. XAFS

Individual Cu species were studied by XAFS in combination with linear combination fitting. Figure 7a shows the X-ray absorption near edge structure (XANES) spectra of Cu K-edge for Cu$_{5.6}$-SAPO-44, Cu$_{1.6}$-SAPO-44-IE, and reference samples. The XANES spectrum of Cu$_{5.6}$-SAPO-44 resembled that of CuO, while that for Cu$_{1.6}$-SAPO-44-IE is similar to that of CuSO$_4$. The corresponding Fourier-transformed k$^3$-weighted Cu K-edge patterns give more direct evidence (Figure 7b). In the case of Cu$_{5.6}$-SAPO-44, two characteristic peaks at approximately 2.6 and 3.0 Å were detected, which are attributed to the neighboring Cu atoms (Cu-O-Cu) in CuO [25]. For Cu$_{1.6}$-SAPO-44-IE, the peak at ~1.5 Å assigned to the Cu-O scatterings is similar with that of CuSO$_4$, confirming the predominance of isolated mononuclear Cu$^{2+}$ species [26]. In addition, weak peaks at ~2.5 Å for Cu$_{1.6}$-SAPO-44-IE were distinguished, indicating the coexistence of few CuO nanoparticles. The relative amount of Cu$^{2+}$ and CuO could be obtained from the intense analysis of XANES spectra using linear combination fitting (LCF) [27]. For Cu$_{5.6}$-SAPO-44, the majority of Cu species are CuO nanoparticles, while the Cu$^{2+}$ ions are dominant in Cu$_{1.6}$-SAPO-44-IE (Figure 8). Thus, the isolated Cu$^{2+}$ proportion was significantly improved via IE.

Consequently, the enhanced high-temperatures activity is attributed to the removal of CuO agglomerations and the relocation of the Cu$^{2+}$ ions for Cu$_x$-SAPO-44-IE [10,17]. However, Cu$_{1.6}$-SAPO-44-IE still exhibited lower activity at high temperature (350–550 °C) compared with the traditional ion-exchanged counterpart with a comparative Cu content that contains only isolated Cu$^{2+}$ ions [28].

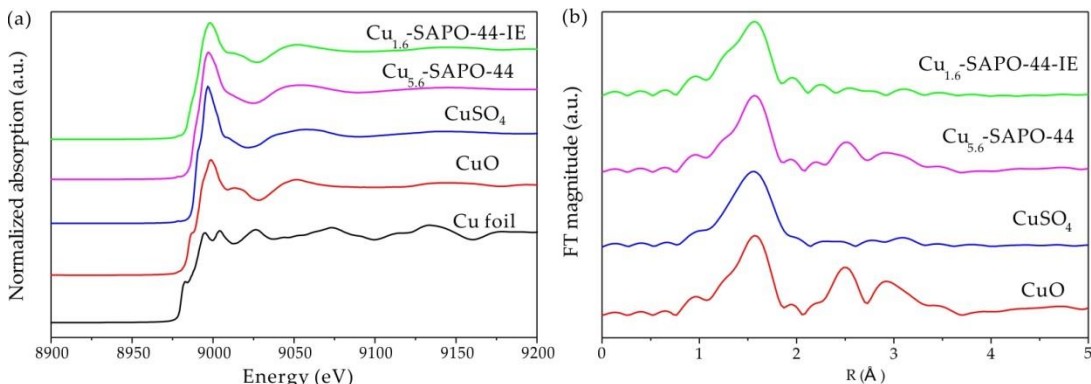

**Figure 7.** XANES (**a**), and FT-EXAFS (**b**) of Cu K-edge spectra for $Cu_{5.6}$-SAPO-44, $Cu_{1.6}$-SAPO-44-IE and the references.

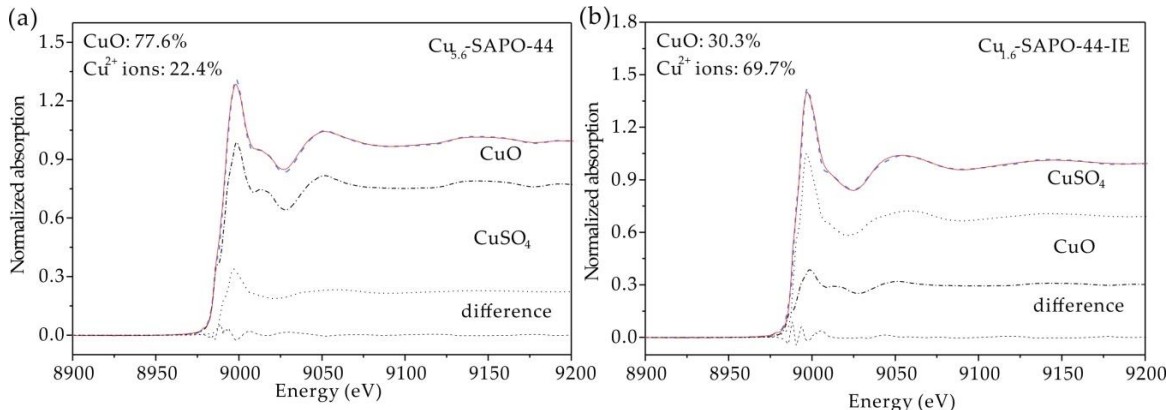

**Figure 8.** Linear combination fitting (LCF) of XANES spectra for (**a**) $Cu_{5.6}$-SAPO-44, (**b**) $Cu_{1.6}$-SAPO-44-IE.

## 3. Materials and Methods

### 3.1. Catalyst Preparation

In preparation of Cu-SAPO-44, the gel has a molar ratio of 0.14–0.22 Cu:0.44 TEPA:0.6 $SiO_2$:0.8 $Al_2O_3$:1 $P_2O_5$:40 $H_2O$:2 N,N,N′,N′-tetramethyl-1,6-hexanediamine (TMHD). $CuSO_4$ solution (20 wt.%) and TEPA were mixed under vigorous stirring at room temperature to get a homogeneous sol After that, phosphoric acid ($H_3PO_4$, 85 wt.% aqueous solution), pseudo-boehmite ($Al_2O_3$, 78 wt.%), colloidal silica sol (30 wt.% suspension in water), TMHD template and deionized water were added to the Cu-TEPA complex solution under vigorous stirring. Then, the resulting viscous gel was aged with stirring at room temperature over 12 h, and further hydrothermally heated statically in a Teflon-lined steel autoclave at 200 °C for 96 h. The product was obtained by centrifugal, washed several times with deionized water, and dried at 100 °C for 12 h. $Cu_x$-SAPO-44 zeolites were obtained by calcination at 550 °C for 6 h in air to remove the structure-directing agent, where "*x*" represents the Cu content in the catalyst determined by ICP analysis.

In this study, the as-prepared Cu-CHA zeolite samples, have high Cu contents of 4.3, 5.6, and 7.6 wt.% respectively. Then unroasted $Cu_x$-SAPO-44 (1 g) zeolites were conducted ion exchange with 1.0 M $NH_4NO_3$ solution at 80 °C for 8 h to remove the abundant Cu, and dried $Cu_x$-SAPO-44-IE (0.9 g) can be obtained. Finally, the dried samples were calcined at 550 °C for 6 h [17] to remove structure-directing agent, and obtained $Cu_x$-SAPO-44-IE, samples with Cu contents of 1.0, 1.6, and 2.2 wt.% (corresponding original *x* = 4.3, 5.6, and 7.6 wt.%, respectively).

*3.2. Catalyst Characterizations*

X-ray diffraction (XRD) was used to obtained the information about the crystalline structure of the catalysts, and equipped with Cu K$\alpha$ radiation ($\lambda$ = 1.5418 Å). The BET surface area and pore size distribution of Cu$_{5.6}$-SAPO-44 and Cu$_{1.6}$-SAPO-44-IE were determined using BET measurements (Micromeritics ASAP 2020, Norcross, GA, USA) after dehydration of the catalysts at 300 °C for 9 h under vacuum. ICP-atomic emission spectrometry (ICP-AES) was carried out to determine the elemental contents of samples by using a PerkinElmer Optima 2100DV (Waltham, MA, USA). FE-SEM equipped with energy dispersive spectroscopy (EDS) was performed on a Hitachi SU-70 microscope (Tokyo, Japan). HRTEM was conducted on a JEOL JEM-2010 (Tokyo, Japan) and a FEI Tecnai G2 F20 transmission electron microscope operating at 200 kV. H$_2$-TPR experiments were performed on a chemisorption analyzer (XianQuan, TP-5000, TianJin, China). The samples (50 mg) were pretreated with pure O$_2$ flow at 500 °C for 30 min, and cooled down to the room temperature in the presence of O$_2$. Subsequently, the samples were reduced under a flow of 5 vol.% H$_2$/N$_2$ (50 mL/min) and then was heated to 800 °C with the rate of 10 °C/min. XAFS spectra were measured for the Cu K-edge at 1W1B beamline of Beijing synchrotron radiation facility (BSRF, Beijing, China) in the transmission mode and fluorescence mode at room temperature. XAFS raw data were analyzed using IFEFFIT software package [29].

*3.3. Catalyst Activity Measurements*

The steady state NH$_3$-SCR activity tests of Cu-SAPO-44 catalysts (~120 mg, 40–60 meshes) were performed at atmospheric pressure in a fixed-bed quartz tube reactor (6.0 mm i.d.). The gas consisting of 500 ppm NO, 500 ppm NH$_3$, 5.3 vol. % O$_2$, 5 vol. % H$_2$O (when used) and balance, He. The total gas flow rate was 300 mL/min, corresponding to a gas hourly space velocity (GHSV) of 100,000 h$^{-1}$. The measurement was carried out in the temperature range 100–550 °C. The NO and NO$_2$ concentrations of the reactor inlet/outlet were monitored by a chemiluminiscence NO$_x$ analyzer (42i-HL, Thermo, Waltham, MA, USA). In addition, N$_2$O and NH$_3$ were detected by quadrupole mass spectrometer (MS, OmniStar 200, Balzers, Switzerland) with a m/z of 44 for N$_2$O, and 17 for NH$_3$. During the test, each temperature point is stable for at least 30 min. The catalytic activity and N$_2$ selectivity were calculated according to the following equations:

$$\text{NO}_x \text{ conversion } (\%) = \frac{[\text{NO}_x]_{\text{in}} - [\text{NO}_x]_{\text{out}}}{[\text{NO}_x]_{\text{in}}} \times 100\%,$$

$$\text{N}_2 \text{ selectivity}(\%) = \frac{[\text{NO}_x]_{\text{in}} + [\text{NH}_3]_{\text{in}} - [\text{NO}_x]_{\text{out}} - [\text{NH}_3]_{\text{out}} - 2[\text{N}_2\text{O}]}{[\text{NO}_x]_{\text{in}} + [\text{NH}_3]_{\text{in}} - [\text{NO}_x]_{\text{out}} - [\text{NH}_3]_{\text{out}}} \times 100\%.$$

## 4. Conclusions

Cu-SAPO-44 zeolites were synthesized using a one-pot approach with dual-amine templates. Furthermore—although the total content of Cu was decreased—the CuO dispersion and isolated Cu$^{2+}$ proportion were significantly improved via the subsequent IE with NH$_4$NO$_3$. The SCR activity was thus promoted at higher temperature. Cu$_{1.6}$-SAPO-44-IE shows >90% NO$_x$ conversion in the temperature range 200–500 °C.

**Author Contributions:** Conceptualization, N.Z., Y.X. and Z.Z.; data curation, N.Z. and Y.X.; formal analysis, N.Z., Y.X., Q.L., and Z.Z.; funding acquisition, Y.X. and Z.Z.; investigation, N.Z., X.M., Y.Q., L.Z. and Y.X.; methodology, N.Z., L.Z. and Y.X; writing–original draft, N.Z. and Y.X.; writing–review and editing, Z.Z.

**Funding:** This research was funded by National Natural Science Foundation of China (Grant No. 21906063 and 21876061) and Key Technology R&D Program of Shandong Province (Grant No. 2019GSF109042).

**Conflicts of Interest:** The authors declare no conflict of interest.

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
