# Peer review of "Ion Exchange of One-Pot Synthesized Cu-SAPO-44 with NH4NO3 to Promote Cu Dispersion and Activity for Selective Catalytic Reduction of NOx with NH3"

_catalysts, doi:10.3390/catal9110882_

Round 1

Reviewer 1 Report

This paper describes the one-pot synthesis of Cu-SAPO-44 and subsequent ion exchange by treatment with NH4NO3. The so synthesized materials show very good properties as robust catalysts in the NOx reduction with NH3. Experimental work is well planned and performed. The results are fully supported by the data and the draft is well written and easily understandable. In my opinion, this work deserves to be published in Catalysts. I got just some minor suggestions in order to improve the manuscript. 

The experimental section (point 3.1) should be improved mainly IE step. The acronyms used must be cleared (TMHD ???). 

The authors must include recovery experiments.

Author Response

Thanks for your kind instruction on our previous manuscript. We have carefully taken your comments into consideration in preparing our revised version. The followings summarized the responses to your comments.

Reviewer #1:

The experimental section (point 3.1) should be improved mainly IE step. The acronyms used must be cleared (TMHD ???).

Reply: As you suggested, ion exchange steps have been improved in the revised manuscript. The full name of TMHD, N,N,N',N'-tetramethyl-1,6-hexanediamine, is supplemented in the revised manuscript.

(Page 7, line 193-198; line 183.)

The authors must include recovery experiments.

Reply: The details of the recovery experiments have been added in the revised manuscript. The results show that the recovery of zeolite after NH4NO3 treatment is ≥90%.

(Page 7, line 194-196.)

Reviewer 2 Report

The manuscript describes the preparation of Cu modified SAPO-44 catalysts, the Cu content of which was decreased by post-synthesis one-pot ion exchange with aqueous ammonium nitrate. This was reported to remove CuO species and to redisperse Cu2+, finally decreasing ammonia oxidation in DeNOx reaction at high temperature and increasing N2 selectivity. The submission is written straightforward and well structured. The applied techniques were appropriate and investigated structural features explain the catalyst performance well. However, in some parts, the discussion needs some clarification (see below). I recommend publishing this paper after minor revisions.

General comments:

The authors should explain the chemical background of preferential removal of CuO species (described as CuO nanoparticles in l. 108). How the ion exchange takes place at atomic level? Which ions are exchanged?

The application of TPR might have been instructive to elucidate nature and redox activity of the Cu sites.

It would have been also nice to see an experiment in presence of water. Significance of such "dry" experiments is somewhat limited.

Here are detailed comments and criticisms:

Results and Discussion

The BET and XRD data for the pure support should be provided as a reference.

Figure 1: The overlapping of text and curves should be avoided to improve the readability.

Table 1: how do the authors interpret the irregular course of Si/Al ratio with increasing Cu content in both series of catalysts? The Si/Al ratios are NOT "nearly the same" as stated in l. 74!

Material and Methods

Catalyst load and particle size should be specified.

The reference state for the gas volumes is not given (standard or normal conditions?).

Conclusions

The statement on improved isolated Cu2+ proportion is imprecise: it increases related to total Cu content, but the mass fraction in catalyst decreases.

Necessary corrections

L. 67: "retain" instead of "remain"

Figure 3b: what is "GSTP" in y axis label?

Ll. 158, 175 and 187: "Catalyst"

Author Response

Thanks to your careful reading and thoughtful comments on our previous manuscript. We have modified the reversion carefully according to your suggestion. The followings are our responses to these comments.

Reviewer #2:

General comments:

The authors should explain the chemical background of preferential removal of CuO species (described as CuO nanoparticles in l. 108). How the ion exchange takes place at atomic level? Which ions are exchanged?

Reply: The chemical background of preferential removal of CuO species have been supplemented in the revised manuscript. According to the cited literature, large CuO particles can be scoured into CuO nanocrystals and re-dispersed as Cu2+ ions during the ion exchange process.

(Page 5, line 117-121.)

The application of TPR might have been instructive to elucidate nature and redox activity of the Cu sites?

Reply: H2-TPR experiments have been supplemented as suggested and the result show that large agglomerated CuO particles can be removed effectively and highly dispersed Cu2+ and CuO nanocrystals were obtained after NH4NO3 treatment. The experimental details and discussion of H2-TPR results are added in the revised manuscript.

(Page 8, line 208-211; page 5-6, line 141-151.)

It would have been also nice to see an experiment in presence of water. Significance of such "dry" experiments is somewhat limited.

Reply: As you suggested, NH3-SCR performance of Cu1.6-SAPO-44-IE catalyst was evaluated in the presence of 5 vol.% H2O in feeding gas. A decrease in NOx conversion was observed at low temperatures (<250oC), while NOx conversion at high temperatures (>400oC) was promoted. The discussion and experimental details have been added in the revised manuscript.

(Page 3-4, line 93-100; page 8, line 218.)

Results and Discussion

The BET and XRD data for the pure support should be provided as a reference.

Reply: BET and XRD data of pure SAPO-44 have been reported in our previous work, which are directly cited in the revised manuscript from reference 18.

(Page 2, line 64-66; page 4, line 103.)

Figure 1: The overlapping of text and curves should be avoided to improve the readability.

Reply: We are sorry for this carelessness. The figure has been modified in the revised manuscript.

(Page 2, line 70-72.)

Table 1: how do the authors interpret the irregular course of Si/Al ratio with increasing Cu content in both series of catalysts? The Si/Al ratios are NOT "nearly the same" as stated in l. 74!

Reply: The discussion on Si/Al ratio for fresh and NH4NO3-treated samples in Table 1 has been revised in the revised manuscript.

(Page 2, line 74-75.)

Material and Methods

Catalyst load and particle size should be specified.

Reply: The catalyst load and particle size have been specified in the revised manuscript as suggested.

(Page 8, line 216.)

The reference state for the gas volumes is not given (standard or normal conditions?).

Reply: The reference state for the gas volumes have been given in the revised manuscript.

(Page 8, line 217.)

Conclusions

The statement on improved isolated Cu2+ proportion is imprecise: it increases related to total Cu content, but the mass fraction in catalyst decreases.

Reply: The expression of conclusion has been improved in the revised manuscript.

(Page 8, line 230.)

Necessary corrections

L. 67: "retain" instead of "remain" Figure 3b: what is "GSTP" in y axis label? Ll. 158, 175 and 187: "Catalyst"

Reply: We are very sorry for these mistakes. The corresponding revisions have been given in the revised manuscript.

(Page 2, line 69; page 4, Figure 4b; page7-8, line 181, 199, 215.)